# Climate change rapidly warms and acidifies Australian estuaries

Elliot Scanes [1✉], Peter R. Scanes [2] & Pauline M. Ross [1]

Climate change is impacting ecosystems worldwide. Estuaries are diverse and important aquatic ecosystems; and yet until now we have lacked information on the response of estuaries to climate change. Here we present data from a twelve-year monitoring program, involving 6200 observations of 166 estuaries along >1100 kilometres of the Australian coastline encompassing all estuary morphologies. Estuary temperatures increased by 2.16 °C on average over 12 years, at a rate of 0.2 °C year$^{-1}$, with waters acidifying at a rate of 0.09 pH units and freshening at 0.086 PSU year$^{-1}$. The response of estuaries to climate change is dependent on their morphology. Lagoons and rivers are warming and acidifying at the fastest rate because of shallow average depths and limited oceanic exchange. The changes measured are an order of magnitude faster than predicted by global ocean and atmospheric models, indicating that existing global models may not be useful to predict change in estuaries.

[1] School of Life and Environmental Sciences, the University of Sydney, Sydney, NSW, Australia. [2] Estuaries and Catchments Science, New South Wales Department of Planning, Industry and Environment, Sydney, NSW, Australia. ✉email: Elliot.scanes@sydney.edu.au

Predicting the response of estuaries to global climate change remains speculative[1,2]. Estuaries worldwide are dynamic ecosystems that vary in morphology, size, catchment and oceanic exchange[3–5]. Large riverine estuaries and bays are warming in North America, e.g. Hudson River[6,7], Chesapeake Bay[8], Woods Hole[9], Narrow River[10] and Europe, North[11] and Mediterranean Seas[12]. While this knowledge is useful, we currently lack detailed information on the impact of global change in a diverse range of morphological estuary types across continental and global regions.

Estuaries link the land and ocean to provide valuable ecosystem services, such as nutrient cycling, carbon storage, trophic transfer, aquaculture and wild fisheries, and create novel habitats vital for biodiversity, and as nurseries for commercially valuable fish species[1–3,13]. Eastern Australia has diverse morphological estuary types interspersed over >3000 km of coastline spanning multiple bioregions[3]. Estuaries typical of subtropical and warm-temperate eastern Australia are generally shallow, and similar to those found in other dry temperate parts of the world like South Africa[14] and the Mediterranean[15]. South-eastern Australia has a consistent semi-diurnal tidal range of 1.2–1.8 m, and the estuaries have previously been categorised into five types (Fig. 1): creek, river, lagoon, lake and back dune lagoon (BDL) based on their capacity to retain catchment inflows without flooding (retention) and the efficiency of tidally driven water-flushing exchange through the entrance (see "Methods")[16,17]. Relatively low rainfall regimes and longshore drift of beach sands due to coastal wave energy cause some estuary types to become periodically closed[3]. Lakes and rivers generally have enough catchment inflow to retain an open connection to the ocean, while lagoons, creeks and BDLs often experience closing of oceanic connections during low flow. All estuary types are sufficiently shallow to remain vertically mixed by wind and tidal energy[3,16].

Estuaries have unique ecological and economic roles. In Australia, the largest estuaries, lakes and rivers, are responsible for wild fishery catch[18] and large-scale processing of nutrient exchange from their catchments[16,18]. These estuaries are also the focus of development and industry, such as ports or housing, and providing places for recreational boating and fishing[3]. The relatively smaller and more shallow creeks, lagoons and BDLs are responsible for cycling nutrients on a local scale and providing nursery habitats for juvenile fish[2,16]. Despite their smaller size, creeks, lagoons and BDLs are more numerous than lakes and rivers and provide a wide range of exposed and shallow-water habitats vital to the functioning of coastal ecosystems[3]. Many of these estuaries contain protected habitats and are important for aquatic and avian biodiversity. They provide critical habitat and feeding grounds for internationally significant migratory shore birds on the Indo-Pacific Flyway and for nationally protected shore and water birds in Ramsar wetland sites[19].

The diversity, complexity and generally small size of Australia's non-tropical estuaries, hinder the use of remote sensing to monitor change[20], and current large-scale oceanic or atmospheric models are limited in their capacity to predict change. Therefore, current ocean and atmosphere modelling provides very little insight into how shallow estuaries are affected by climatic warming, either on Australia's eastern coast or worldwide. We present the results from a 12-year monitoring study investigating the summer temperature, pH and salinity of 166 estuaries along >1100 km of the Australian temperate to subtropical east coastline, encompassing the full range of estuary morphologies present, and including inter-yearly cycles of Australian weather drivers (i.e. Southern Oscillation Index and Indian Ocean Dipole).

To understand estuary change, a "Random forest" model was trained on the collected data to predict estuary temperature, pH and salinity using the attributes of estuaries as predictor variables (Table 1). By identifying the processes and attributes responsible for estuary change, new larger-scale models may be able to be created, so change can be more accurately predicted in estuaries around the globe.

## Results

**Temperature**. Overall, estuaries have warmed by 2.16 °C in the last 12 years (0.2 °C year$^{-1}$). While all estuary types, except creeks, showed a significant warming trend over the 12-year study period, there were differences among estuary types. Lagoons warmed the most with temperature increasing in lagoons over the last 12 years by up to 3.65 °C (0.325 °C year$^{-1}$). Rivers were the second fastest and warmed by 0.248 °C year$^{-1}$, followed by Back Dune Lagoons (BDLs, 0.117 °C year$^{-1}$) and lakes (0.0954 °C year$^{-1}$) that warmed at a similar rate to each other (Table 2).

Random forest (RF) models confirmed that estuary temperature has steadily increased over the last 12 years (Supplementary Fig. 1). The strongest driver of temperature was time of measurement, January and February being the hottest months. Time measured as days since sampling began was the next strongest driver followed by latitude of the estuary (Fig. 2a). Retention time, flushing time and average depth were identified as the strongest drivers of estuary temperature after time (days and months) and latitude (Table 3, Fig. 2a). As the average depth of estuaries decreased, temperature increased (Fig. 3). This effect of estuary depth becomes stronger over the study period, having a relatively larger influence in the second half of the study period. Partial dependence plots revealed that the greatest warming occurred when flushing time was long, average depths were shallow and total volumes were small to intermediate (Fig. 3).

**Estuary pH**. All estuaries were found to be acidifying, with lagoons and creeks acidifying the fastest and lakes the slowest (Table 2). Despite the shorter time of measurement, pH of estuaries was measured for 6 of the 12 years of the monitoring programme; there was a significant decrease in pH of ~0.098 units year$^{-1}$ (Fig. 1, Table 2). RF models confirmed that the month of sampling, latitude and days since sampling began were the most important predictors of pH (Table 3, Fig. 2b). pH steadily declined over the 6 years of sampling, but the rate of decline increased as latitude (distance from the equator) increased (Supplementary Fig. 2). RF models identified seagrass cover and increase in nitrogen load as drivers of pH (Fig. 2b). Generally, estuaries that acidified the least contained the most seagrass.

**Salinity**. Estuaries had highly variable salinity over the sampling period although there was a slight trend for an overall decrease (Fig. 1, Table 2), this was also reflected in RF models (Supplementary Fig. 3). Change in salinity was dependant on estuary type; creeks and lagoons became less saline, rivers increased in salinity and there were no significant changes in lakes or BDLs. RF models confirmed that days elapsed, month of measurement and then retention factor of the estuary were the three most important predictor variables (Fig. 2c, Table 3). Australian Bureau of Meteorology climate data[21] shows that over the past decade, rainfall has decreased. This has, somewhat counter-intuitively, led to creeks and lagoons becoming significantly less saline as they fill with freshwater runoff while entrances are closed, while rivers were more saline due to oceanic intrusion as a consequence of lower relative freshwater input.

## Discussion

Across 166 Australian estuaries over the last 12 years, the summer estuary water temperature increased on average by 2.16 °C

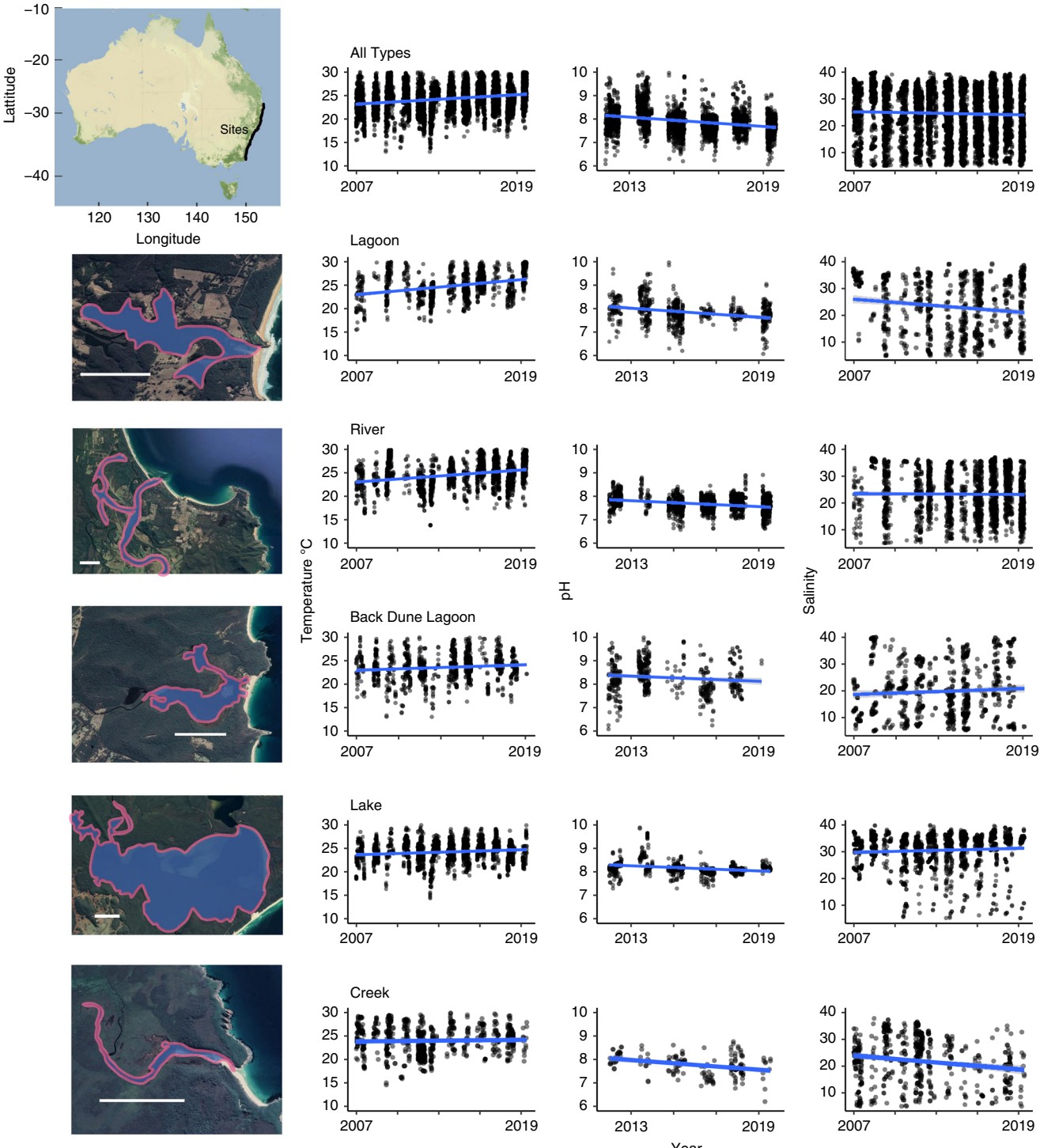

**Fig. 1 Observed change in estuary temperature, pH and salinity since 2007.** Summer temperature pH and salinity measured over the 12- (temperature and salinity) and 6-year (pH) estuary monitoring programme from December 2007 to January 2019; grey dots indicate each data point, darker dots indicate multiple data on that point. Estuaries are divided into the five estuary types; each estuary is represented by a satellite image of an estuary that is typical of the type, with the body of the estuary outlined in pink. White scale bar indicates 1000 m. Satellite images of estuaries are sourced from Google Earth (map data: SIO, NOAA, US Navy, NGA and GEBCO). All estuaries are represented on a map of Australia showing the sample sites as black dots. The map of Australia is sourced from Stamen Design, under CC BY 3.0. Data by OpenStreetMap, under ODbL.

(0.2 °C year$^{-1}$), pH decreased by ~0.5 pH units and salinity decreased slightly (mean −0.97, −0.086 PSU year$^{-1}$). Australian air and sea temperatures have increased by ~1 °C since 1910[21]; however, eastern Australia is warming faster than the rest of the continent. Over the last decade, summer air temperatures in

eastern Australia have generally increased by 1.5 °C, and sea surface temperatures have increased by ~1 °C compared with the 1961–1990 average[20]. Our findings show that overall, estuaries are warming generally faster than these temperatures, with much of the warming concentrated in lagoons and rivers, with less

**Table 1 Predictor variables used to determine the drivers of change in temperature, pH and salinity in east Australian estuaries.**

| Predictor variable | Conceptual link and source of evidence | Type of predictor | Appropriate for analysis |
|---|---|---|---|
| Days elapsed | Time since monitoring began[28,57] | Time | Temperature, pH and salinity |
| Month of measurement | Intra-annual variability[28] | Time | Temperature, pH and salinity |
| Retention factor | Ratio of estuary potential total volume to run-off volume[17] | Geomorphology | Temperature, pH and salinity |
| Latitude of site (°S) | Regional climate cline[3,28] | Geomorphology | Temperature, pH and salinity |
| Size of the catchment (km$^2$) | Approximate freshwater input[3] | Geomorphology | Salinity |
| Average depth (m) | Radiative heat exchange[2,22,79] | Geomorphology | Temperature, pH and salinity |
| Total flush time of the estuary (days) | Seawater exchange[3,17] | Geomorphology | Temperature, pH and salinity |
| Percentage of the estuary area covered by seagrass | Photosynthetic activity[34] | Geomorphology | pH |
| Percent of catchment cleared (%) | Stream and catchment shading and heating of overland flow over cleared landscapes[22] | Human disturbance | Temperature, pH and salinity |
| Percentage of the catchment urbanised | Urban heat[22] | Human disturbance | Temperature |
| Proportional increase in nitrogen load | Changes to catchment land use[17,57] | Human disturbance | Temperature, pH and salinity |

**Table 2 Results of simple linear models for temperature, pH and salinity over time categorised by estuary type.**

| | Creek | River | Lake | Lagoon | BDL | ALL |
|---|---|---|---|---|---|---|
| **Temperature (°C)** | | | | | | |
| Coefficient | −0.00017 | 0.00068 | 0.000262 | 0.00089 | 0.000319 | 0.00052 |
| *p* value | NS | <0.0001 | <0.0001 | <0.0001 | <0.001 | <0.0001 |
| $R^2$ (adjusted) | 0.0019 | 0.082 | 0.0149 | 0.103 | 0.0114 | 0.0466 |
| Number of observations (*n*) | 625 | 1892 | 1751 | 1054 | 949 | 6271 |
| Change in °C year$^{-1}$ | NS | 0.248 | 0.0954 | 0.325 | 0.117 | 0.192 |
| Change over the sampling period (°C) | NS | 2.79 | 1.07 | 3.65 | 1.31 | 2.16 |
| **pH** | | | | | | |
| Coefficient | −0.000276 | −0.000168 | −0.000146 | −0.000243 | −0.000166 | −0.000237 |
| *p* value | <0.001 | <0.0001 | <0.001 | <0.0001 | <0.01 | <0.0001 |
| $R^2$ (adjusted) | 0.0325 | 0.0623 | 0.00608 | 0.0586 | 0.0128 | 0.0846 |
| Number of observations (*n*) | 122 | 1132 | 644 | 598 | 641 | 3137 |
| Change in pH units year$^{-1}$ | −0.101 | −0.0612 | −0.0534 | −0.0888 | −0.0607 | −0.0978 |
| Change over the sampling period | −0.53 | −0.32 | −0.28 | −0.46 | −0.32 | −0.51 |
| **Salinity (PSU)** | | | | | | |
| Coefficient | −0.00164 | 0.000653 | −0.000277 | −0.00131 | 0.000242 | −0.000238 |
| *p* value | <0.0001 | <0.001 | NS | <0.0001 | NS | <0.05 |
| $R^2$ (adjusted) | 0.032 | 0.006 | 0.0003 | 0.016 | −0.0006 | 0.0005 |
| Number of observations (*n*) | 625 | 1892 | 1751 | 1054 | 949 | 6271 |
| Change in PSU year$^{-1}$ | −0.6 | 0.238 | NS | −0.479 | NS | −0.0861 |
| Change over the sampling period (PSU) | −6.74 | 2.68 | NS | −5.38 | NS | −0.97 |

change in lakes and creeks. Previous work has shown that small inland creeks[22] and large riverine estuaries[6,8,9] follow the trend of air temperature. Our results suggest that this may hold true for some estuaries; however, the drivers of change are complex, and the factors affecting the degree of warming in estuaries are more than just air or ocean temperature.

Average depths of estuaries have previously been shown to play a large role in the temperature of rivers and lakes in the northern hemisphere[22]. In this study, the warmest estuaries had flushing times between 30 and 120 days. Random forest models confirmed that after time of measurement, latitude, average depth and flushing time of estuaries were important drivers of estuary temperature. Shallower average depths provide a greater capacity for the estuary to absorb radiative heat per volume of water.

Flushing time is affected by entrance condition (whether the estuary was open to the ocean or blocked by sand) and overall volume. A morphological warm spot was apparent when flushing time was long enough for retained water to be warmed by radiative heat; average depths were shallow, and total volumes were small to intermediate.

Lagoons were the fastest warming estuary type. Lagoons fall within the "warm spot" of shallow average depths and short–medium flushing times. Conversely, lakes are the coolest because of their relatively greater depths (generally still less than 10 m), and longer flushing times. Average depth and flushing times are not the only variables controlling temperature, but explain much of the difference observed among estuary types. Many of the estuaries known to be warming from previous

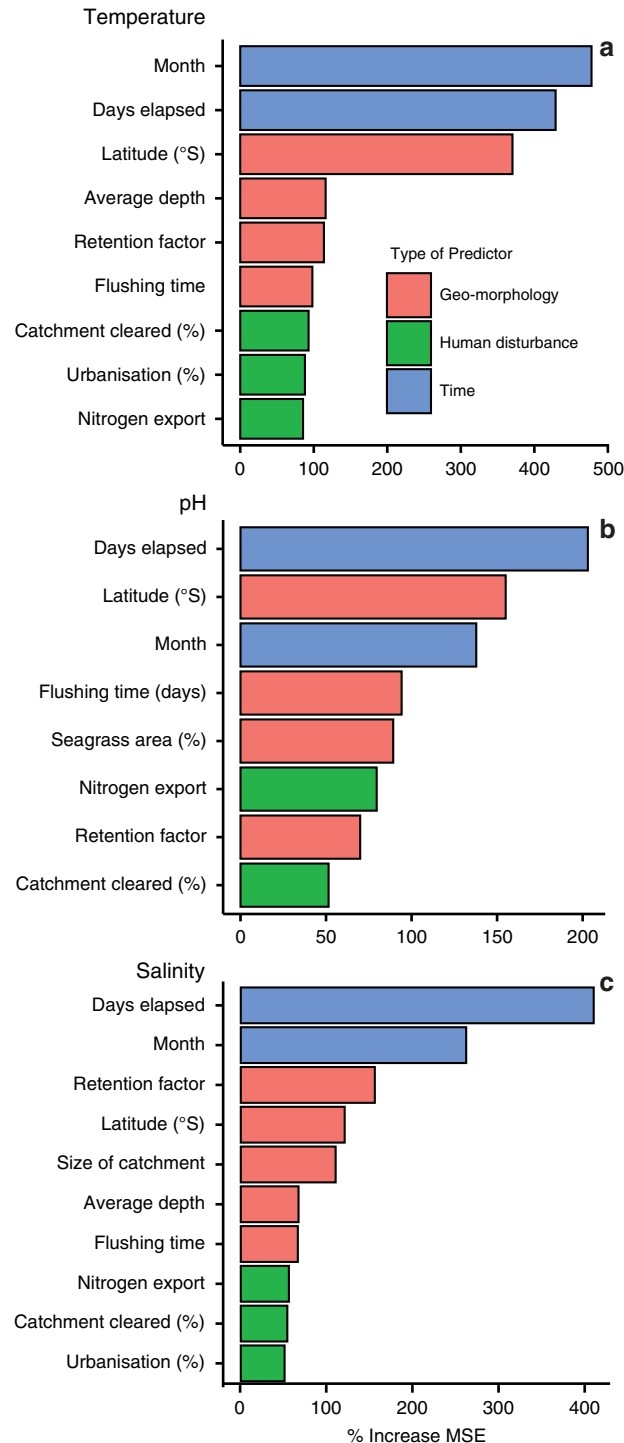

**Fig. 2 Relative importance of variables in predicting estuary temperature, pH and salinity.** Variable importance plots (% MSE as an indicator of importance[73,74]) for **a** temperature, **b** pH and **c** salinity generated from random forest models. Predictor variable categories are colour coded.

**Table 3 Model validation metrics.**

| Model | % Variance explained[a] | RMSE[a] | RMSE[b] | $R^{2\,b}$ | MAE[b] | p value[c] |
|---|---|---|---|---|---|---|
| Temperature | 82.59 | 1.223 | 1.344 | 0.794 | 0.934 | <0.001 |
| pH | 81.02 | 0.238 | 0.251 | 0.793 | 0.164 | <0.001 |
| Salinity | 85.65 | 4.345 | 4.616 | 0.839 | 2.95 | <0.001 |

Models were tested using a 20% withhold 10-fold cross-validation technique and the randomisation technique recommended by Murphy et al.[76] (see Methods for details). RMSE root-mean square error, MAE mean absolute error.
[a]Calculated from the "out-of-the-bag" predictions[74].
[b]Calculated using 20% withhold 10-fold cross-validation[73].
[c]Calculated using the randomisation technique[76].

The "heat island effect" resulting from removal of riparian vegetation decreases the amount of shade over water[23], and an increase in hot-paved areas warms the water that flows into estuaries[24]. Rising river temperatures in Europe, North America and Asia have been attributed to heated wastewater and large paved areas[22,25].

Creeks had the greatest average temperature, but showed no significant warming over the sampling period. Creeks heat quickly because of their shallow depths and medium-to-low flush time, but they are also prone to frequent displacement-opening to oceans where all water is exchanged in a short period (low retention)[3]. We suggest that the shallow average depths allow rapid warming, but the low retention of creeks does not allow water to continue to gain heat over time. Creeks are the most variable estuary type, but were under-represented in this sampling programme. Large inherent variability and lower sampling effort reduces our potential to capture the change in the temperature of creeks.

pH decreased by 0.5 units over the 6 years that pH was measured. This decline is faster than that projected by IPCC (0.00625 pH units year$^{-1}$, pH 7.7) for the year 2100 under "business as usual" emission scenarios for the open ocean[26]. pH is dependent on temperature and should decrease as temperature increases; however, the magnitude of pH increase as seen in this study far exceeds that expected by the effects of temperature on pH[27]. These pH changes also cannot be explained by increased rainfall as Eastern Australia has dried over the last decade[21]. While $pCO_2$ changes in the atmosphere contribute to pH change in estuaries, these dynamic systems are influenced by complex interactions between catchment characteristics, cover of submerged vegetation and season[27]. These factors are partially correlated with greater air temperatures, less seagrass cover and greater summer rainfall in lower latitudes (northern coast). The rainfall trend is identified to become more pronounced due to climate change into the coming century[28]. Decadal pH recordings in Neuse River estuary, USA show pH declines of ~0.02 units year$^{-1}$. Those declines were dependent on $pCO_2$ concentrations, catchment conditions and primary production[27]. Van Dam and Wang[27] suggest that the acidification of the Neuse River estuary is largely dependent on localised conditions. Our data show wide-scale acidification of all estuary types, suggesting that in addition to local influences, acidification processes are pervasive and are not confined to individual estuaries.

RF models identified seagrass cover and increase in nitrogen load as drivers of pH (Fig. 2b). Estuaries that acidified the least contained the most seagrass. Seagrasses and algae photosynthesise during the day using nutrients and $CO_2$ that in turn increase pH. pH increased where seagrass cover and nitrogen load (which is linked to pelagic algal biomass[29]) increased. Longer flushing and retention times increased pH, perhaps because of greater pelagic primary production in retained water bodies[30]. Seagrasses are declining worldwide, with up to one-third of seagrasses lost globally[31]. Among eight of the estuaries in this study, which were

studies are doing so at slower rates than reported here[6–8]. This is likely because they are large, deep and tidally driven, e.g. the Hudson River[6,7], Chesapeake Bay[8] compared with the smaller lagoon-type estuaries monitored in this study.

Estuary temperatures were shown to be less dependant on catchment characteristics than attributes of the estuaries. Estuaries were, however, warmer when catchments were urbanised.

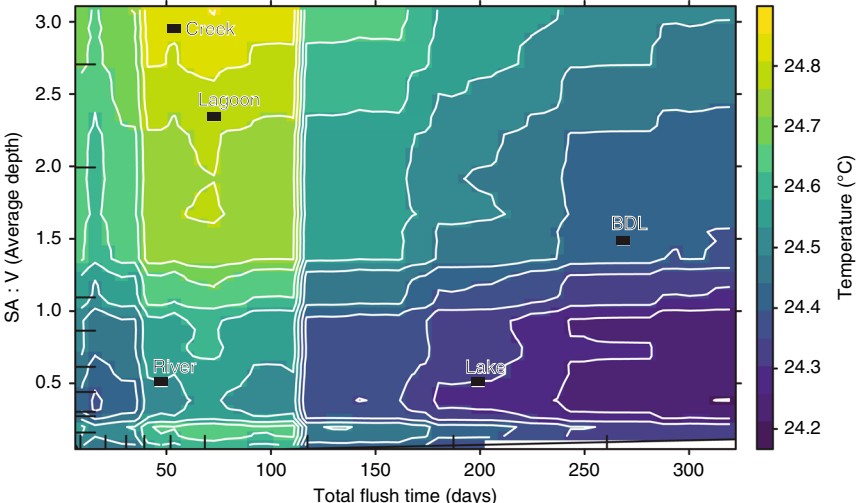

**Fig. 3 Estuary temperature modelled using average depth and flushing time as predictors.** Temperature as modelled by RF over a range of average depths and flushing times. Each estuary type is shown using its mean average depth and flushing time to indicate their general location in relation to these predictors.

included in a global survey of seagrass loss, seven declined in seagrass area at an average rate of 0.8% year$^{-1}$ [31]. Seagrass loss is expected to accelerate as the oceans warm and heatwaves become more common[32,33]. The degree of warming observed in this study could negatively affect seagrasses by reducing their capacity to sequester $CO_2$, causing estuarine acidification to accelerate in the coming decades[32,33]. Maintaining seagrasses and other photosynthesisers will be crucial to mitigate estuarine acidification in the coming century[34].

Estuaries had highly variable salinity over the sampling period although there was a slight trend for an overall freshening or decrease (Fig. 1, Table 2). During the period of sampling, however, eastern Australia experienced ~10% lower-than-average rainfall[21]. Lower rainfall reduces the inflow to estuaries, which results in longer periods of entrance closure in intermittent estuaries and greater saline intrusion into open estuaries[3,35]. In our analysis, creeks and lagoons became significantly less saline as they slowly fill with freshwater runoff while entrances are closed, while rivers with open entrances became more saline from lower relative freshwater input and greater oceanic intrusion. Greater retention times and shallower average depths of estuaries initially result in lower salinity from runoff, but eventually greater salinity from evaporation[35]. Increases in salinity have been observed in some closed estuaries in 2019 and 2020, but those data are not included in this analysis. RF models showed that salinity increased in southern estuaries as latitude (distance from the equator) increased, which may again be a consequence of greater summer rainfall in northern regions[21]. These patterns will continue as rainfall in eastern Australia becomes more sporadic and delivers larger volumes in isolated rainfall events[21]. In addition, predicted sea-level rise will increase the prevalence of coastal flooding and storm surges that will alter salinity dynamics[2]. Predictor variables dependent on human impacts (i.e. percent of catchment cleared, increase in nitrogen load and urbanisation) were significant, but ranked lower in importance. As the percentage of catchment cleared and nitrogen load increased, there was a general decrease in salinity perhaps because of the greater capacity for runoff due to less catchment vegetation[36].

The changes observed in this study have occurred against a background of altering climate conditions in Australia. Since 1970, rainfall in southern Australian has decreased, with a

concurrent decline in streamflow, including the South East Coastal drainage basin, where this study is focused[21]. Low streamflow can contribute to warming of estuaries by increasing their retention times and decreasing average depth[37]. Such low-flow conditions have the greatest effect in smaller shallow-water bodies like lagoons and rivers. The combination of lower streamflow and increased air temperature may begin to explain the large increased temperatures in lagoons and rivers as seen in this study.

Inter-yearly weather drivers can play a significant role in decadal temperature trends. In Australia, the major drivers of weather patterns are the Southern Oscillation index (SOI, responsible for El Niño and La Niña events) and the Indian Ocean Dipole (IOD)[21]. The Australian Bureau of Meteorology has stated that inter-yearly variability in Australian weather caused by the SOI and IOD is occurring against the backdrop of a general warming trend, rather than the SOI and IOD driving the warming trends themselves[21]. The multiple environmental variables presented here have been measured for longer than a decade, including sampling across different inter-yearly cycles encompassing three periods of El Niño (2007, 2010 and 2016), three periods of La Niña (2009, 2011 and 2017–2018)[38] and both positve (2012 and 2015) and negative (2010, 2014 and 2016) IOD events[39]. By sampling over multiple inter-yearly drivers, these data avoid confounding and bias towards either SOI- or IOD-dominated weather patterns, and capture the general trends of temperature, pH and salinity. Australian winter temperatures have followed the same warming pattern as summer temperatures over recent decades[21].

This study has shown that eastern Australian estuaries are warming, acidifying and freshening more quickly than predicted by global models for the air or oceans[40]. Estuaries are highly diverse and complex systems that make accurate models by predicting general estuary change difficult to create. Our results highlight that air or ocean temperatures alone cannot be relied upon to estimate climate change in estuaries; rather, individual traits of any estuary need to be considered in the context of regional climate trends. Shallow estuaries with periodically closing entrances, typical of eastern Australia, are the most vulnerable to warming, and deeper, large estuaries are least vulnerable. Such shallow estuaries with medium flushing times are common

worldwide, especially in other dry temperate zones such as Western Australia[2], South Africa[14] and the Mediterranean[15]. Warming and drying of temperate areas are occurring worldwide, with examples in southern Europe[41,42], South Africa[43] and California[44]. The similarity between these regions and eastern Australia means that patterns found in this study are relevant for estuaries in temperate zones worldwide.

The rapid climate change is likely to impact marine and estuarine organisms and ecosystems. Thresholds or "tipping points" of ecosystems are difficult to establish, especially in complex and diverse environments like estuaries where all feed-back loops and interactions cannot be accounted[45]. We do, however, know that warming and acidification can cause greater energetic demand, altered behaviour, altered morphology, development and lower fecundity in a range of estuarine and marine organisms, including plants[31,46], invertebrates[33,47] and vertebrates[48–52]. These effects are known to be greater when multiple stressors combine such as acidification, warming and salinity[49,53]. Already, reports of range shifts towards the poles[54] of mobile marine species have been observed. Coastal environments and temperate ecosystems in eastern Australia and other warming hotspots around the globe are beginning to tropicalise[55,56]. Estuaries that remain open may also soon begin to tropicalise, and estuarine ecosystems become colonised by tropical marine species and reflect a warmer environment[2]. Although such range shifts and tropicalisation present new opportunities for tourism, fisheries and aquaculture, this will require stakeholders to invest that will not happen quickly and will require a transition period[48].

Estuaries that periodically close like creeks, lagoons and BDLs may respond differently and not tropicalise. It has been suggested that they will become more saline under future conditions as entrances remain closed for longer periods of time, allowing for greater evaporation[2]. We found that despite lower streamflow conditions, these estuaries freshened, perhaps because of freshwater inflows while closed. Freshening estuaries may experience lower species richness, as ecosystems shift towards those dominated by freshwater species, and closed entrances exclude marine species and their larvae from colonising estuaries[1,2]. Low-flow conditions in closed estuaries can increase the risk of algal blooms and hypoxia that will impact sensitive taxa like small crustaceans and juvenile fish[2,57], ultimately lowering species richness. It is, however, likely that this trend will change to salinisation if rainfall further decreases and flows become too small to compensate for evaporation[2].

Estuaries provide services of immense ecological and economic value. The rates of change observed in this study may also jeopardise the viability of coastal vegetation like mangroves and saltmarsh in the coming decades[33] and reduce their capacity to mitigate storm damage and sea-level rise[58]. Changes in temperature, pH and salinity are likely to reduce the global profitability of aquaculture[33,59] and wild fisheries[60]. Aquaculture and fisheries are worth USD 70 million[18] to NSW estuaries; however, globally 56.9 million people rely on these industries for income[61]. Global aquaculture is worth USD 243.5 billion, and wild fisheries USD 152 billion, a large proportion of which occurs in estuaries[61]. The impact of climate change on these industries will be felt strongest in regions that rely on aquaculture and fisheries in shallow estuaries for culture, income and food. This is of concern in other dry temperate zones like the Mediterranean and South Africa where many of the estuaries are similar to those studied here.

Human activities within catchments can affect the temperature and pH of estuaries; we found that increased cleared areas caused estuaries to warm, and greater seagrass area can buffer against acidification. Estuaries around the globe are centres of industry

and urbanisation where human activities introduce nutrients[62], pollutants[63], invasive species[64] and artificial structures[65] while extracting resources[61]. These activities are likely to magnify the effects of climate change on estuarine ecosystems, reducing human impacts in estuaries and catchments that will be vital to relive pressures on the most vulnerable estuaries as climate change intensifies.

Existing regional-scale climate change modelling necessarily uses large grid cells and broad-scale response variables to create generalised outcomes across a region. Homogeneity is assumed within grid cells for most variables, tacitly ignoring small-scale variation within grid cells (e.g. related to elevation or land use or estuaries). This study has focussed on estuaries, which are an important ecosystem from both an ecological and a cultural perspective. Changes in water-quality variables, such as temperature, pH and salinity, can critically reshape estuarine ecosystems; yet, how these variables in estuaries are affected by climate change is poorly represented by regional ocean models. Existing studies that measure the response of estuaries to climate change have focussed on detailed studies of single sites (e.g. Hudson River[6,7], Chesapeake Bay[8], Woods Hole[9], Narrow River[10] and Europe, North[11] and Mediterranean Seas[12]). Whilst valuable for each system, these studies are of limited use for regional-scale models due to their narrow focus and no understanding of the validity of generalising findings to other estuaries. In contrast, our study provides an understanding of how a range of variables (e.g. estuary type, average depth, macrophyte abundance and catchment disturbance) interact with climate change to influence response at large spatial scales and over many estuaries. This study provides a detailed understanding of the factors that influence climate outcomes in shallow estuaries, and the data demonstrate that changes may be occurring at rates faster than those predicted by regional ocean or atmosphere models. These outputs provide the foundational understanding to improve models used to determine the impacts of climate change on ecosystems and human communities in coastal areas.

In summary, estuaries across 1100 km, encompassing multiple bioregions in southeast Australia, are rapidly warming, acidifying and freshening, at rates greater than those predicted by air or oceanic models. Smaller and shallower estuaries with enclosed entrances and longer retention times are warming and acidifying most rapidly, indicating that climate change is already impacting estuaries on a continental scale. This greater understanding of change in estuaries will enhance regional-scale modelling and allow for informed mitigation in the future.

## Methods

**Sampling.** We present a synthesis of environmental data from 166 estuaries, including 6200 observations over a latitudinal range from 28°S to 37.4°S (~1100 km) encompassing all estuary types (47 Rivers, 43 Lagoons, 28 Lakes, 25 Creeks and 23 Back-dune Lagoons (BDLs)). Estuaries were sampled every summer over a 12-year period (between 2007 and 2019) in the state of New South Wales (NSW) in southeast Australia (Fig. 1). Simple linear models were used to determine the rate of change in temperature, pH and salinity over the study period. Drivers of patterns in response variables (temperature, pH and salinity) were determined using catchment data, geomorphological data and time of sampling as predictor variables to create validated random forest models (see Methods). We tested the prediction that response variables in estuaries would change over time, and the magnitude of change would depend on latitude, geomorphology, hydrology and catchment disturbance.

Sampling was part of a larger estuary health-sampling programme[66,67] and took place in the state of New South Wales (NSW), Australia, over a latitudinal range from 28°S to 37.4°S (~1100 km).

In each austral summer since 2007–2008 to 2018–2019, water parameters at ~0.5-m depth were measured 6 times at each of 2–3 central basin sites in estuaries in NSW. Previous work has shown that due to the relative shallowness, water columns are fully mixed by wave and tidal energy; this means that samples collected at 0.5-m depth are indicative of the entire water column[17]. The "summer" period was defined as November to March. In each summer, data are collected at ~30–40 estuaries from a single region. The region represents approximately one-

**Table 4 Generalised characteristics of NSW estuary types.**

| Estuary entrance group[8] | Metatype[19] | Type | Number sampled | Description |
|---|---|---|---|---|
| Tide dominated | River | Drowned river valley (classed with barrier river) | 5 | River with deep-wide entrance with no impediment to ocean exchange. Large-to- moderate dilution capacity and moderate flushing. |
| Wave dominated open | River | Barrier river | 47 | River with wave-dominated entrance and reduced flushing capacity. Moderate dilution and rapid flushing. The entrance may close occasionally. |
| Wave dominated intermittent | Lake and lagoon | Lake | 28 | Large non-linear water body with substantially restricted entrance. Moderate-to-large dilution capacity but very long flushing time. The entrance may close occasionally. |
| | Lake and lagoon | Lagoon | 43 | Medium-sized non-linear water body with substantially restricted and periodically closed entrance. Moderate dilution capacity but long flushing time. |
| | Creek | Creek | 25 | Small linear or non-linear water body with substantially restricted and periodically closed entrance. Very low dilution capacity and short-to-moderate flushing time. |
| | Lake and lagoon | Back Dune Lagoon | 23 | Medium-sized non-linear water body with substantially restricted and periodically closed entrance. Moderate dilution capacity and long flushing time. Groundwater dependent. |

Estuary types based on Scanes et al.[4] and Roper et al.[17]. Sampling in this study was stratified by metatype, but the findings are reported by type.

third of the coast (North, Central and South). In addition, data were collected at sentinel sites in central and south regions every year. This results in a focus on each region every third year, with a lower level of sampling in other regions. Temperature (±0.001 °C), pH (±0.01) and salinity (±0.01 PSU) are measured at ~0.5-m depth using a calibrated water-quality sonde (YSI EXO 2, Yellow Springs Instruments, Ohio, USA). A sample occurs when ambient water from 0.5-m depth water is pumped through a flow cell on the sonde and data logged every second for 3–5 min. The vessel is free to drift during the sample. The data used in this study are the mean values for each logging period. Samples were haphazardly collected with respect to time, so the sample for an estuary can be at any time of the day. Therefore, the data show variability due to time of collection and variation from the beginning to the end of the summer period, as well as variation due to weather. Sampling occurred across different inter-yearly cycles: three periods of El Niño (2007, 2010 and 2016), three periods of La Niña (2009, 2011 and 2017–2018)[38] and both positve (2012 and 2015) and negative (2010, 2014 and 2016) IOD events[39], avoiding bias towards either SOI- or IOD-dominated weather patterns.

Sampling locations for any year were stratified by estuary type[3,17] (and see below) and by catchment disturbance (see below), and were randomly selected from the pool of possible estuaries within a region.

**Estuary typology.** There are 184 recognised estuaries in NSW. Unlike many temperate areas, particularly in the northern hemisphere, most estuary entrances in Australia are dominated by relatively (by temperate standards) low rainfall regimes and longshore drift of coastal sands, resulting in a dominance of periodically closed estuaries[3,68]. The estuary typology used here is based on a functional classification that uses two criteria[17]: (I) Retention Factor is the ratio of estuary potential total volume to run-off volume. It is a measure of the propensity to open, and is based on the ratio of the capacity for additional water volume within the estuary compared with additional volume resulting from an event producing 10% of the average annual runoff. (II) Flushing Time is the average water turn-over time using the ratio of volume exchanged within a tidal prism method (considering entrance condition) to the overall volume of the estuary.

This has resulted in three estuary "meta-types" and 6 "types" (Table 4). Sampling was stratified by metatype, but analyses in this paper are reported by type (with the exception that no distinction is made between barrier rivers and drowned river valleys, and bays were not sampled).

South-eastern Australia has a latitudinally consistent semi-diurnal open-water tidal range ranging from 1.2 to 1.8 m[69]. This means that there is no latitudinal confounding of flushing time—which is affected by tidal range. The open-water tidal range is attenuated within estuaries, and the degree of attenuation is determined by entrance characteristics.

**Input variables.** Input variables that relied upon models and measurements of characteristics (excludes days elapsed, month of measurement and latitude of site) are described below. These methods were developed as part of a larger estuary and catchment-monitoring programme described in detail in Roper et al.[19].

Retention was conceived as the capacity of an estuary to retain incoming freshwater flows, and associated nutrients and suspended sediments, during a rainfall event without additional discharge to the ocean (flood). A retention factor was calculated as the ratio of the estuary volume to the volume of runoff from a large rainfall event, here defined as an event that results in 10% of the total annual inflow[17]. As an example, a factor of 15 would indicate that the estuary volume is 15 times larger than the runoff from a large rainfall event.

Estuary volumes were determined from hydrographic bathymetry surveys that are available for 57 of the estuaries. These were gridded into a digital elevation model with horizontal resolution of 12.5 m, which was merged with a topographic

DEM of coastal catchments to create a continuous surface up to at least 1.6 m above Australian Height Datum (AHD—approximately mean tidal height)—which is above the high-tide level. Hypsometry was calculated for each and the volume was calculated[16,17].

Volumes for the remaining 127 estuaries were estimated using surface-area-to-volume regressions for the known estuaries. The best regressions were obtained when estuaries were grouped into types according to the typology of Roy et al.[3]. Best-fit regression lines were all exponential functions with $r^2$ ranging from 0.911 to 0.996. The estuaries without hydrographic bathymetry surveys were then grouped by type[3] and the regression for that type was used to estimate the volume from surface area for each estuary[16,17]. Catchment areas were determined from existing national Digital Elevation Models (DEM) data using GIS software (Arc GIS)[16,17,70].

The surface area of each estuary was hand digitised from the NSW 1:25,000 topographic map series. This was then combined with the DEM derived for volume to produce a surface area at 0.6-m AHD[19]. Marine macrophyte mapping (see seagrass area) was superimposed to allow calculation of open-water areas as well as surface areas including emergent macrophytes (mangroves and saltmarsh)[16]. Open-water areas were used in this analysis.

Areas of seagrass were taken from Creese et al.[71] and Roper et al.[16] plus other unpublished data from the authors of the Creese report (T. Glasby) and an author of this paper (P. Scanes). Those data were used to confirm the absence of seagrass from systems where it had not been mapped.

The surface-area-to-volume ratio (SA:V) was determined by dividing the estuary surface area in m[2] (as calculated above) by the volume in m[3] (as calculated above) for each estuary.

The average depth of estuaries was calculated from hypsometry where available (see volume above) or by dividing volume (m[3]) by surface area (m[2]).

Flushing time was calculated directly for all those estuaries with tidal gauging, which included estuaries that are periodically closed. The tidal prism method was used for estuaries with an open, or mostly open, entrance, adopting a typical exchange efficiency coefficient of 0.15.

For those periodically closed estuaries without tidal gauging, the concept of a flushing time required different treatment. For periodically closed estuaries that are closed or mostly closed, a flushing time can be approximated by the period the estuary remains closed[16]. Full details can be found in Roper et al.[16].

To determine the percentage of catchment cleared or urbanised, land-use maps of NSW were used. Land-use mapping for Eastern and Central NSW commenced in April 2001 and was completed in June 2007 by NSW Government. Aerial photography and satellite imagery were acquired between 1999 and 2006 depending on availability and the timing of mapping[16]. Comparison to recent imagery shows that there has not been significant change in land use (at a state scale) since the 2007 mapping was completed.

Land use was mapped to a total of 128 classes using the Australian Land Use and Management (ALUM) scheme and then aggregated into 21 categories for the purposes of hydrological modelling. This was followed by further aggregation into nine classes of native forest, cleared land, urban, crops, grazing, irrigated pasture, dry forb, irrigated forb and others, for the purposes of nutrient and sediment export modelling. Data for all nine land uses excluding native forest were summed to provide an estimate of the total area of disturbed land within each estuary catchment[16].

GIS software (Arc GIS) was used to calculate the area of each aggregated land-use class within the catchment of each of the 166 estuaries.

Proportional increases in nitrogen input loads were calculated to estimate how land-use change since European colonisation of Australia has altered nutrient export into estuaries. A catchment export model was used to calculate nutrient export, sediment export and surface flows for each of the NSW estuary catchments.

The model had a hydrologic flow component that accounted for surface and groundwater flow[16,72]. The model also accounted for land slope, soil type and typical rainfall intensity[16].

To estimate nutrient and sediment exports from each catchment, the freshwater surface flows were multiplied by measured (median) nutrient concentration data obtained from the published scientific literature or from past government-monitoring projects[16,17]. This multiplication produced annual nutrient load exports for each catchment[16]. Concentration data are expressed as event mean concentration (EMC) for each land use, which is equivalent to the mean concentration of nutrients in runoff from a rain event. Median values were used in calculations.

A proportional increase in nitrogen load was calculated by running the model for each catchment with all land use set to native forest (pre-1770 condition), and then with current land use. The ratio of pre-1770:current land-use loads becomes the proportional increase in nitrogen load for each estuary catchment. Approximately 20% of estuary catchments had a ratio <1.2, indicating that their loads are very close to pre-1770 conditions.

**Data analysis.** All data analyses were done using R version 3.5.3 statistical software. Simple linear models were used to determine the rate at which all estuaries, and each estuary type had changed in respect to temperature, salinity and pH over the study period. These methods have been used previously in studies like Kaushal et al.[22]. Model fit and error normality was examined using Pearson's residual plots. Coefficients were then used to determine the modelled change per year and change over the sampling period.

To explore the impact of temporal and environmental variables on temperature, pH and salinity in estuaries, "random forest" supervised machine-learning models were used. Such data-mining techniques allow us to accurately predict and explore mechanistic relationships for large complex data where traditional modelling approaches would be hindered by collinearity, non-independence and non-normality[73]. Three models were created using the "randomForest" package[74], one each for temperature, salinity and pH. Each model used the predictor variables in Table 1 to train the model. Predictor variables were selected for environmental relevance to each response variable. Random forest models are robust to collinearity[74]; however, in cases where there was obvious dependency between variables, only one variable was used based on relevance from the literature[73]. Each variable selected as a predictor in the random forest model was based on published research known to affect the response variable. Random forests were grown using 1000 trees, with each tree using a bootstrap sample of 66% of the data. The number of variables tested at each split (mtry) was set at mtry = 9 for temperature and salinity, mtry = 8 for pH; this was determined using the "tune rf" function[74]. Models were validated by using a 20% withhold 10-fold cross-validation technique as recommended by Evans et al.[73]. Using this data-withhold method, we were able to generate root-mean-square error (RMSE), $R^2$ and mean absolute error (MEA) using the "Caret" package[75] (Table 3). While random forest models are robust to errors, there is still the chance that a given model is no different to random chance. To test this, we used the randomisation technique recommended by Murphy et al.[76]. Briefly, the response variable (temperature, pH and salinity) for each model was randomised and the model was then run and the error (% variance explained) tabulated. This process was repeated 1000 times to generate a distribution of model error on random data. The % variance explained from the real (non-random data) model was then compared with the error distribution for the random model and a p value was created (see ref. [73] for full details).

To determine the importance of input variables in our RF models, we used the % change in model error when a variable was removed[73,74]. The increase in mean square error (% MSE) upon removing variables provides a measure of how much the predictive ability of the model is reduced when the effect of a certain variable is excluded. This is a common method of determining variable importance[73,74]. Partial dependence plots were created using the "Plotmo"[77] and "PDP"[78] packages to view the modelled outputs for each variable (Supplementary Fig. 1).

## Data availability
The data that support the findings of this study are available in the NSW government SEED public database with the identifier https://datasets.seed.nsw.gov.au/dataset/nsw-estuary-temperature-ph-and-salinity-data.

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

## Acknowledgements

The authors wish to acknowledge the employees of New South Wales Department of Planning, Industry and Environment for collecting these data over the last 12 years. We thank Thomas Bishop for suggestions and assistance with statistical analyses.

## Author contributions

P.R.S. designed the monitoring programme and oversaw the collection of data; E.S. analysed data and wrote the paper with P.R.S. and P.M.R.

## Competing interests

The authors declare no competing interests.
