## [Peer Review File · Nature Communications]

Reviewers' comments:

Reviewer #1 (Remarks to the Author):

This study shows that Eastern Australian estuaries in a dry-temperate climate have warmed, acidified and freshened (in some cases) during the last twelve years. These are very important, clear and novel findings. However, I think the authors could have gone some way further in discussing the implications of their finding in a national and global sense. Therefore, I don't recommend publication in nature communications in its present form. My comments are below.

Introduction:

I'm unconvinced that a twelve year monitoring programme is long enough to have confidence in the changes presented. While I'm sure the trends are real, its the rates of change that are surely contingent on variability in the drivers that is not captured within the sampling programme. For example, is the average rise in temperatures of 2.16 degrees Celsius meaningful? i.e. Does your sampling programme capture inter-annual (all types of El Nino, La Nina, and other years), seasonal, and extreme event-scale variability in all drivers (rivers, air temps, storm surges) - in order to correctly calculate the average? And if the variability across time scales and estuary types and latitudes is large, is the quoted average a useful metric?

Why do global models under-predict the changes that you find? This is important to clarify because you clearly state this in your introduction - and models hold obvious and powerful advantages for future impact assessments. However, you don't really discuss this further in your paper, which makes me question why such a clear statement was made in the first place.

Introduction: The statement "...lagoons, followed by rivers,.." is a little confusing - could this definition be written clearer?

Also not clear how urbanisation is accelerating the warming - needs clarification.

Main section:

I felt there needed to be a stronger justification of the importance of these estuary types, within both national and global contexts. For example, their importance for ecology, water cycling/quality, socio-economics, pollution, etc. You do not mention that they are shallow types until the round-up at then end. You say at the end that they provide ecosystem services but this seems a very open-ended statement, with only one follow-on example in the next sentence (you also here relate coastal vegetation (undescribed term) to storms and sea-level rise (undescribed processes), and do not mention e.g sea grasses and carbon capture that have previously been discussed).

When discussing trends in air/sea temperatures, salinities and pH (line 58 onwards), it would have been very useful to have discussed changes in other drivers, notably rainfall and river flows (and changes in the behaviours of river flows and loads). But also changes in land use and water management. You say later that Eastern Australia has dried over the last decade, with sporadic/isolated rainfall events, but by how much? How does this affect rivers? And temperatures? And is there evidence of changes in river intensities and frequencies and combination events (e.g. high-rivers and surge-tides)? Sea-level rise and storms are only briefly mentioned at the end, with no quantification. I think its crucial in these very dynamic systems to get across the variability of the drivers, and the uncertainty in the data.

It was not clear from the article what the implications are of rising temperatures, salinities and pH. Why is this a problem and what are the tolerances and tipping points for different communities and impacts?

When discussing the economic value of estuaries and inferred negative impact of changes in estuaries to this economy, there are presumably global regions where the impact is less and even opportunities for new industries. These ideas could be discussed.

Again, there is limited discussion on the impact of the changes to ecosystems. For example, habitat and species migration, adaption potential of species, vulnerable species near tolerance boundaries, tipping points, invasive species, water quality or pollution, viruses and die-off rates affecting humans and food supply, etc...

Finally, your summary paragraph mentions new concepts such as land-use intensification, then discusses some drivers (sea grasses and urban paving), but not others, and finishes without a clear recommendation for mitigation ("proactive steps" and "reducing human impacts" are quite vague statements).

Reviewer #2 (Remarks to the Author):

Dear Authors;

Last week I read with great interest your work on climate change impact on estuarine water quality parameters. The work reads fluently and to the point and I think you generally draw the right conclusions based on your methodology. The work is very relevant while the large dataset and its analysis is convincing and clear.

Attached I send you my comments in the PDF text. Generally I have the following major comments/questions:

- 1) How would you classify the influence of tidal range in your outcomes? Is that constant along the measured coastal transect? It is somehow reflected in the flushing time. It is an important parameter since it is not only related to volume exchange but also to vertical mixing of warmer inland water to colder sea water. Commenting on the importance of tidal range would help to make the conclusions more transferable to other regions.
- 2) related to that : I learned from figure1 that the seasonally varying parameter variation is quite strong but that the small different estuary trends and parameter responses are still clear. I was initially somehow surprised that your work is based on only 5 minute sampling in many different estuaries at many different points in time (instead of continuous monitoring in all estuaries, which is nearly impossible to finance), but I see that the data and the analysis are quite consistent. I just wondered what would be the impact of measuring (eg temperature) at only 0.5 depth? Would that reflect temperature dynamics in a deep lake or river? Can it be that you are drawing biased conclusions for deeper estuary types where temperature stratification maybe pronounced?
- 3) What about winter temperatures? Do you expect a similar trend?
- 4) I experienced that the Random Forest model applied in the study was not easily understandable from the manuscript alone. I leave it to the Editor whether or not more attention should be paid to this in the final work.

Reviewers comments and authors replies, outlining action taken

Please find below our responses to the reviewer comments. Care has been taken to ensure all comments were addressed.

Reviewer #1 (Remarks to the Author):

This study shows that Eastern Australian estuaries in a dry-temperate climate have warmed, acidified and freshened (in some cases) during the last twelve years. These are very important, clear and novel findings. However, I think the authors could have gone some way further in discussing the implications of their finding in a national and global sense. Therefore, I don't recommend publication in nature communications in its present form. My comments are below.

Introduction:

Reviewer's comments	Authors replies
I'm unconvinced that a twelve year monitoring programme is long enough to have confidence in the changes presented. While I'm sure the trends are real, its the rates of change that are surely contingent on variability in the drivers that is not captured within the sampling programme. For example, is the average rise in temperatures of 2.16 degrees Celsius meaningful? i.e. Does your sampling programme capture inter-annual (all types of El Nino, La Nina, and other years), seasonal, and extreme event-scale variability in all drivers (rivers, air temps, storm surges) - in order to correctly calculate the average? And if the variability across time scales and estuary types and latitudes is large, is the quoted average a useful metric?	This is the only dataset which measures over the long-term, multiple environmental variables of estuarine systems. Being greater than a decade long, sampling included different inter-yearly cycles; across three periods of El Niño (2007, 2010, 2016), three periods of La Niña (2009, 2011, 2017-2018) ³⁸ and both positive (2012, 2015) and negative (2010, 2014, 2016) IOD events ³⁹ ; avoiding confounding and bias towards either SOI or IOD dominated weather patterns. The Australian Bureau of Meteorology has stated that the "The year-to-year changes in Australia's climate are mostly associated with natural climate variability such as El Niño and La Niña in the tropical Pacific Ocean and phases of the Indian Ocean Dipole in the Indian Ocean. This natural variability now occurs on top of the warming trend, which can modify the impact of these natural drivers on the Australian climate". By spanning multiple inter-yearly drivers, this data captures the general trends of temperature, pH and salinity. This represents climate related change consistent with the trend of warming in the air and ocean. On L 205 - 216 in the discussion we have inserted an entire paragraph to discuss this issue and provide evidence that our study does capture all inter-yearly weather drivers. The average of 2.16 °C for all estuaries is used to compare estuarine change to global ocean and atmospheric models, or long-term trends over the large area of eastern Australia. These atmospheric models use data from a wide geographical range that masks local variability. We agree that averages for estuary types are more useful to describe the changes in estuaries and predict future changes. The discussion has now been refocused to highlight the importance of considering estuary types. we are confident that the

	sampling period has thus captured the average trends in Australian weather.
Why do global models under-predict the changes that you find? This is important to clarify because you clearly state this in your introduction - and models hold obvious and powerful advantages for future impact assessments. However, you don't really discuss this further in your paper, which makes me question why such a clear statement was made in the first place.	Global models for the air and ocean underestimate the change in estuaries because they do not account for the geomorphology of estuaries and were not designed to model estuarine change (no model exists for this to our knowledge). This has now been discussed on L219-230 of the discussion. The following has been inserted into the discussion on L 219 "This study has shown that eastern Australian estuaries are warming, acidifying and freshening more quickly than predicted by global models for the air or oceans⁴⁰. Estuaries are highly diverse and complex systems making it difficult to create models which predict estuary change. Our results highlight that air or ocean temperatures cannot be relied upon to estimate estuary change; but rather the individual traits of any estuary need to be considered in the context of regional climate trends." And on L 275 "Existing regional-scale climate change modelling necessarily uses large grid cells and broad scale response variables to create generalised outcomes across a region. Homogeneity is assumed within grid cells for most variables, tacitly ignoring small-scale variation within grid cells related to estuaries, elevation or landuse. This study has focussed on estuaries, which are an important ecosystem from both an ecological and a cultural perspective. Changes in water quality variables such as temperature, pH and salinity can critically reshape estuarine ecosystems, yet how these variables in estuaries are affected by climate change is poorly represented by regional ocean models. Existing studies which measure the response of estuaries to climate change have focussed on detailed studies of single sites (e.g Hudson River^{6, 7}, Chesapeake Bay⁸, Woods Hole⁹, Narrow River¹⁰, and Europe; North¹¹ and Mediterranean Seas¹²). Whilst valuable for each system, these studies are of limited use for regional scale models due to their narrow focus and no understanding of the validity of generalising findings to other estuaries. In contrast, our study provides an understanding of how a range of variables (e.g. estuary type, average depth, macrophyte abundance, catchment disturbance) interact with climate change to influence response at large spatial scales and over many estuaries. This study provides a detailed understanding of the factors that influence climate outcomes in shallow estuaries and the data demonstrate that changes may be occurring at rates faster than predicted by regional

	ocean or atmosphere models. These outputs provide the foundational understanding to improve models used to determine the impacts of climate change on ecosystems and human communities in coastal areas.”
Introduction: The statement "...lagoons, followed by rivers,.." is a little confusing - could this definition be written clearer?	This has now been rephrased to read “Importantly, the response of estuaries to climate change is dependent on their morphology; for example, lagoons and rivers, are warming and acidifying at the fastest rate due to high surface area to volume ratios and limited oceanic exchange.”
Also not clear how urbanisation is accelerating the warming - needs clarification	This has been addressed in the revised discussion. Please see L 141 in the revised manuscript. The following has now been inserted “Estuaries were, however, warmer when catchments were urbanised. The “heat island effect” resulting from removal of riparian vegetation decreases the amount of shade over water ²³ and an increase in hot paved areas warms the water which flows into estuaries ²⁴ . Rising river temperatures in Europe, North America and Asia have been attributed to heated wastewater and large paved areas ^{22, 25} . “

Main section:

I felt there needed to be a stronger justification of the importance of these estuary types, within both national and global contexts. For example, their importance for ecology, water cycling/quality, socio-economics, pollution, etc. You do not mention that they are shallow types until the round-up at then end. You say at the end that they provide ecosystem services but this seems a very open-ended statement, with only one follow-on example in the next sentence (you also here relate coastal vegetation (undescribed term) to storms and sea-level rise (undescribed processes), and do not mention e.g sea grasses and carbon capture that have previously been discussed).	An entire paragraph has now been inserted in the introduction to address this issue. Please see lines L 50 -60 in the revised introduction. The following has been inserted into the manuscript: “Estuaries have unique ecological and economic roles. In Australia, the largest estuaries; lakes and rivers, are responsible for wild fisheries catch ¹⁸ and large-scale processing of nutrient exchange from their catchments ^{16, 18} . These estuaries are also the focus of development and industry such as ports or housing; and providing places for recreational boating and fishing ³ . The relatively smaller and more shallow creeks, lagoons and BDLs are responsible for cycling nutrients on a local scale and providing nursery habitats for juvenile fish ^{2, 16} . Despite their smaller size, creeks, lagoons and BDLs are more numerous than lakes and rivers and provide a wide range of exposed and shallow-water habitats vital to the functioning of coastal ecosystems ³ . Many of these estuaries contain protected habitats and are important for aquatic and avian biodiversity. They provide critical habitat and feeding grounds for internationally significant migratory shore birds on the Indo-Pacific Flyway and for nationally protected shore and water birds in Ramsar wetland sites ¹⁹ .”
--	---

When discussing trends in air/sea temperatures, salinities and pH (line 58 onwards), it would have been very useful to have discussed changes in other drivers, notably rainfall and river flows (and changes in the behaviours of river flows and loads). But also changes in land use and water management. You say later that Eastern Australia has dried over the last decade, with sporadic/isolated rainfall events, but by how much? How does this affect rivers? And temperatures? And is there evidence of changes in river intensities and frequencies and combination events (e.g. high-rivers and surge-tides)? Sea-level rise and storms are only briefly mentioned at the end, with no quantification. I think its crucial in these very dynamic systems to get across the variability of the drivers, and the uncertainty in the data.	We have now elaborated on these points and changes have been made to the discussion to address these issues. Please see L 200-206 for discussion relating to river flows, how they have changed and how this affects our study. The following has been inserted into the manuscript: “The changes observed in this study have occurred against a background of altering climate conditions in Australia. Since 1970, rainfall in southern Australian has decreased, with a concurrent decline in streamflow, including the South East Coastal drainage basin, where this study is focused²¹. Low streamflow can contribute to warming of estuaries by increasing their retention times and decreasing average depth³⁷. Such low flow conditions have the greatest effect in smaller shallow water bodies like lagoons and rivers. The combination of lower stream flow and increased air temperature may begin to explain the large increased temperatures in lagoons and rivers as seen in this study.” And on L 194 “In addition, predicted sea level rise will increase the prevalence of coastal flooding and storm surges which will alter salinity dynamics²”
It was not clear from the article what the implications are of rising temperatures, salinities and pH. Why is this a problem and what are the tolerances and tipping points for different communities and impacts?	The biological and ecosystem effects have now been discussed on L 231-273 of the revised discussion. We have created a new subheading “Consequences for Marine and Estuarine Organisms and Ecosystems ” that focuses on explaining how this change could impact estuarine ecosystems, estuarine and marine organisms and human communities. This study did not measure any biological variables, therefore we feel it might be unwise to make too many assertions regarding the effects of the observed change on marine organisms and ecosystems. We do draw on literature which has independently investigated the effects on estuarine and marine organisms to strengthen this section.
When discussing the economic value of estuaries and inferred negative impact of changes in estuaries to this economy, there are presumably global regions where the impact is less and even opportunities for new industries. These ideas could be discussed.	While there may be some benefits on estuarine change, unfortunately the evidence suggests that negative effects will outweigh most positives. The following has been inserted under the “Consequences for Marine and Estuarine Organisms and Ecosystems ” “ Although such range shifts and tropicalisation present new opportunities for tourism, fisheries and aquaculture, this will require stakeholders to invest which will not happen quickly and will require a transition period⁴⁸.”
Again, there is limited discussion on the impact of the	Please see above comment, we have significantly

changes to ecosystems. For example, habitat and species migration, adaption potential of species, vulnerable species near tolerance boundaries, tipping points, invasive species, water quality or pollution, viruses and die-off rates affecting humans and food supply, etc...	increased our discussion on the biological and ecosystem effects of the observed changes in estuaries with a new subheading and section in the discussion.
Finally, your summary paragraph mentions new concepts such as land-use intensification, then discusses some drivers (sea grasses and urban paving), but not others, and finishes without a clear recommendation for mitigation ("proactive steps" and "reducing human impacts" are quite vague statements).	The concluding paragraph has been re-written without the phrases mentioned by the reviewer. We have ensured no new information is introduced in the concluding paragraph. Please see the revised manuscript.

Reviewer #2 (Remarks to the Author):

Dear Authors;

Last week I read with great interest your work on climate change impact on estuarine water quality parameters. The work reads fluently and to the point and I think you generally draw the right conclusions based on your methodology. The work is very relevant while the large dataset and its analysis is convincing and clear.

Attached I send you my comments in the PDF text. Generally I have the following major comments/questions:

1) How would you classify the influence of tidal range in your outcomes? Is that constant along the measured coastal transect? It is somehow reflected in the flushing time. It is an important parameter since it is not only related to volume exchange but also to vertical mixing of warmer inland water to colder sea water. Commenting on the importance of tidal range would help to make the conclusions more transferable to other regions.	South-eastern Australia has a very consistent tidal range across the entire latitudinal span of this study. We have inserted the following on L 41 in the introduction "South-eastern Australia has a consistent semi-diurnal tidal range of 1.2-1.8 m and the estuaries have previously been categorised into five types..." And L 348 in the methods "South-eastern Australia has a latitudinally consistent semi-diurnal open-water tidal range ranging from 1.2 - 1.8 m. This means that there is no latitudinal change to flushing time – and confounding which might be affected by alterations in tidal range. The open-water tidal range is attenuated within estuaries and the degree of attenuation is determined by entrance characteristics."
2) related to that : I learned from figure1 that the seasonally varying parameter variation is quite strong but that the small different estuary trends and parameter responses are still clear. I was initially somehow surprised that your work is based on only 5 minute sampling in many different estuaries at many different points in time (instead of continuous monitoring in all estuaries, which is nearly impossible to finance), but I see that the data and the analysis are quite consistent. I just wondered what would be the impact of measuring (eg temperature) at only 0.5 depth? Would that reflect temperature dynamics in a deep lake or river? Can it be that you are drawing biased conclusions for deeper estuary types where temperature stratification maybe pronounced?	Previous work has shown that due to the overall shallow nature of Australian Estuaries, water columns are generally well mixed. We have inserted the following on L 315 in the methods: "Previous work has shown that due to the relative shallowness, water columns are fully mixed by wave and tidal energy; this means that samples collected at 0.5m depth are indicative of the entire water column¹⁷."
3) What about winter temperatures? Do you expect a similar trend?	Winter temperatures would likely show a similar trend albeit, with lower upper temperature values. Climate data has shown that warming and rainfall trends are occurring on top of inter-yearly and seasonal variation. The following has been inserted on L 217 in the discussion "Australian winter air temperatures have followed the same warming pattern as summer air temperatures over recent decades²⁰."
4) I experienced that the Random Forest model	This has now been addressed with changes to the

applied in the study was not easily understandable from the manuscript alone. I leave it to the Editor whether or not more attention should be paid to this in the final work.	methods section, and the manuscript is now reformatted. Please see L 449-453 in the revised manuscript. We have inserted the following to help with the explanation of this analysis: “To explore the impact of temporal and environmental variables on temperature, pH and salinity in estuaries, “random forest” supervised machine learning models were used. Such data mining techniques allow us to accurately predict and explore mechanistic relationships for large complex data where traditional modelling approaches would be hindered by collinearity, non-independence, and non-normality⁷³”.
--	--

Extra comments and replies from PDF

L23 over what period?	This text has now been revised to read “We find that estuary temperatures have increased by 2.16°C on average over the last 12 years, at a rate of 0.2°C year⁻¹, with waters acidifying at a rate of 0.09 pH units and freshening at 0.086 PSU year⁻¹.”
L73 somehow I like "average depth" better; it is shorter and more clear to understand	This change has been made throughout the revised version.
L79 There are potentially two effects by radiation: 1) the incoming water from the catchment is warmer 2) the estuary water is warmed faster. It does not become clear to me which one is important for which estuary type; is your methodology able to discriminate between these two processes? pls discuss	Our modelling showed that the average depth of estuaries was ranked quite high as an important driver of temperature (irrespective of type); while, catchment characteristics such as land clearing and urbanisation were ranked low. This indicates that estuary water is warming while in the estuary, rather than in the catchment. This has been further discussed on L 123-139 in the revised manuscript.
L97 Does this hold for all estuary types?	Yes – in the analysis we used the continuous variables that define estuary types, rather than the types themselves. This has now been covered further in the discussion and L 335 -353 in the methods.
L150 Another characteristic of your dataset is that all estuaries face a similar tidal range (or not?). Pls discuss. if not, sensitivity should be apparent from the flushing time (as you define it). In that case it would be an independent input variable worthwhile exploring. Flushing time itself is also a function of estuarine volume.	From the comment above: South-eastern Australia has a very consistent tidal range across the entire latitudinal span of this study. We have inserted the following in L 350-353 in the methods “South-eastern Australia has a latitudinally consistent semi-diurnal open-water tidal range ranging from 1.2 - 1.8 m. This means that there is no latitudinal confounding of flushing time – which is affected by tidal range. The open-water tidal range is attenuated within estuaries and the degree of attenuation is determined by entrance characteristics.”
L324 Calling this an "error" is somewhat confusing	This has now been addressed in the revised methods

although it may be appropriate in a statistical sense. The way you use it reads as if a larger "error" points to a more important driver. This is intuitively difficult to understand.	section. Text has been revised to read on L 469 “To determine the importance of input variables in our RF models, we used the % change in model error when a variable was removed^{47, 48}. The increase in Mean Square Error (%MSE) upon removing variables provides a measure of how much the predictive ability of the model is reduced when the effect of a certain variable is excluded. This is a common method of determining variable importance^{47, 48}.”
L556 But what is the difference between gray and black dots in the graphs?	All dots are grey – the darker dots occur when multiple data points have fallen on the spot in the graph. This has been clarified by adding “Temperature pH and salinity measured over the 12 year (temperature, salinity) and six year (pH) estuary monitoring program; grey dots indicate each data point, darker dots indicates multiple data on that point.” to the figure 1 legend.
L 585 Maybe I missed it , but it would be nice to include the nr of estuaries per type as well.	This has now been included for each estuary type in the revised version of Table 4.
L588 MSE reads as an error whereas you men governing factor (see also earlier comment).	This has now been addressed in a revised methods section. Please see L 471 -476 in the revised manuscript.
Figure 2D: It would be better to describe areas instead of points for different estuary types	The points are there to provide only an indication of the general attributes for the estuary; the figure caption has been revised. We believe that adding error bars to these points would complicate the figure, when estuaries are plotted as a guide only.

REVIEWERS' COMMENTS:

Reviewer #1 (Remarks to the Author):

The manuscript is much improved.
I'm happy that you have addressed all my comments.
I have no further issues with the manuscript.

REVIEWERS' COMMENTS:

Reviewer #1 (Remarks to the Author):

The manuscript is much improved.
I'm happy that you have addressed all my comments.
I have no further issues with the manuscript.

Authors reply:

We thank the reviewer for the helpful comments and feedback.